# Barriers and Facilitators to Conversation: A Qualitative Exploration of the Experiences of People with Parkinson’s and Their Close Communication Partners

**DOI:** 10.3390/brainsci12070944

**Published:** 2022-07-19

**Authors:** Karen Wylie, Hayley M. Carrier, Andreas M. Loftus, Ramishka Thilakaratne, Naomi Cocks

**Affiliations:** 1School of Allied Health, Curtin University, P.O. Box U1987, Perth, WA 6845, Australia; andrea.loftus@curtin.edu.au (A.M.L.); ramishka.thilakaratne@curtin.edu.au (R.T.); naomi.cocks@curtin.edu.au (N.C.); 2School of Biomedical and Allied Health Sciences, University of Ghana, Legon, Accra P.O. Box LG 25, Ghana; 3Wize Therapy, 177–179 Davy Street, Booragoon, WA 6154, Australia; hayley_carrier@hotmail.com; 4Rocky Bay, P.O. Box 53, Mosman Park, WA 6912, Australia

**Keywords:** Parkinson’s disease, communication, intervention, lived experience, communication partner

## Abstract

Conversations are an important part of our daily lives, enabling us to interact with others and participate in a range of activities. For people with Parkinson’s, conversation can be challenging when communication is impacted. This qualitative exploratory study investigated the experiences of people with Parkinson’s and their close communication partners in conversations. The study explored influences on conversational participation, considering perceived barriers and facilitators to participation in conversation for people with Parkinson’s. Interviews were conducted with eight dyads, with participants interviewed both jointly and separately (24 interviews). Five themes revealed that conversation appears to be influenced not only by the communication skills of the person with Parkinson’s, but also by factors associated with the communication partner, the complex nature of conversations, the communication environment, and the impact of experience in shaping participation in conversation. Specific barriers and facilitators to conversational participation were identified. This study offers important insight into the lived experience of people with Parkinson’s affected by communication difficulties. The findings support the notion that it is more than simply the communication skills of the person with Parkinson’s that influence conversations. It is important that other factors influencing conversational success should be included in interventions supporting communication for people with Parkinson’s.

## 1. Introduction

Conversations involve a communicative exchange between two or more people. Conversations enable an individual to engage meaningfully with others and are important in enabling participation in social, household and employment roles. Conversely, difficulty participating in conversation can contribute to social isolation, relationship breakdown and challenges in employment [1]. Each participant in a conversation plays an important role in initiating, maintaining, and concluding conversations [2,3]. The communication skills of each participant, as well as external factors, such as background noise, can impact conversation [2,4,5,6]. If one participant in the conversation has a communication impairment, then conversation is likely to be even more impacted [7,8]. Similarly, the ability of the communication partner (CP), the other person participating in the conversation, to adapt their communication when conversation breakdown occurs also influences conversational success [3,5]. 

Approximately 90% of people with Parkinson’s (PwP) experience communication impairment, which varies in presentation and severity [9,10]. Communication impairment may include difficulties with speech, voice, language, social communication, or a combination of these difficulties [11,12,13,14,15]. All of these difficulties impact the person with Parkinson’s (the PwP) ability to converse [16].

As communication success is dependent on both the CP and the PwP’s communication skills, it is surprising that traditional therapy approaches targeting communication for PwP have focussed on speech production and included only the PwP (e.g., Lee Silverman Voice Treatment) [8]. One therapy approach that has included PwP and their CPs is called Communication Partner Training (CPT) [17]. In this approach, conversations between the CP and the PwP are analysed to observe specific difficulties. Following this, both parties are given strategies to produce more successful conversations. While CPT has been widely used as an intervention for people with other types of communication difficulties, such as aphasia [17], there has been limited research which has explored its use with PwP and their CPs [3]. While CPT offers promise for PwP and their CPs, before we begin to design therapy approaches in this domain, it is important to understand more about the issues PwPs and their CPs experience while having conversations.

There are two main approaches harnessing knowledge about conversation for PwP and their CPs. Observational studies directly analyse conversational exchanges between PwP and their CPs using conversation analysis [11,18,19,20]. These studies describe ways in which people respond to conversation breakdown, including several communication strategies and methods of conversational repair employed by the CP and the PwP. Whitworth et al. [20] described responses to communication breakdown as facilitatory (e.g., encouraging topic maintenance or referencing a new topic), acceptance (e.g., moving from topic to topic without attempting to pick up any common link), confrontational (e.g., directly confronting the PwP by saying ‘Do you know what I’ve said?’ in the case of a delayed response), avoidance (e.g., ignoring a behaviour or comment, avoiding conversation), and emotive (e.g., yelling at PwP when they failed to take a turn during conversations or reporting annoyance in the absence of a response). For example, repair strategies could include requesting for clarification/modification, providing a guess, suggestion, elaboration or specification. While conversation analysis studies provide important insights regarding observable conversational skills of both participants, they are limited because only observed behaviour can be analysed [11,18,19,20]. As such, they do not give information about the participant’s lived experience of conversation. Furthermore, while one is able to hypothesise about how observed communication skills may impact participation in conversations, other contextual factors limiting or facilitating conversational success are not able to be explicitly explored, such as the influence of different environments or communication partners. 

For most people, there are individuals with whom they more frequently converse, such as a spouse, family member or roommate, considered to be a close communication partner (CCP) who has a wealth of experience communicating with the PwP. Qualitative research can be used to explore the lived experience of PwP and their communication partners in conversations. One method that can be used to explore the barriers and facilitators to conversation participation is talking directly to PwP and their CCPs to gather their perspectives about their experiences, including what they perceive to be barriers and facilitators to everyday conversations. Three studies have used interviews to explore barriers and facilitators to conversational participation. Whitehead [15] interviewed four PwP and three of their CCPs about the effect of communication changes they had experienced as a result of Parkinson’s. While not specifically about conversational participation, this study gave insight into some barriers to conversation experienced by PwP. PwP described how their communication difficulties impacted their ability to take part in conversations and frequently led to them being excluded from conversations. While there was some discussion of generic strategies to improve communication, there was no specific consideration of how to improve participation in conversation.

Miller et al. [16] described the impact of communication challenges faced by PwP on their lives. While again not specifically about participating in conversations, some resultant themes described barriers and facilitators to conversation participation. PwP clearly indicated that they experienced difficulties with conversations, linked to both their communication and cognitive difficulties [16]. Specifically, they described difficulties getting into conversations and holding their place in conversations. PwP also spoke about how others frequently talked over them, did not wait for answers, ignored them and did not appreciate the difficulties that they were having with communication. Facilitation strategies involved both the PwP and their CP. PwP discussed that they frequently chose not to speak unless directly addressed or used non-verbal methods to communicate rather than conversations [16]. PwP discussed how CPs supported them to overcome challenges in conversations by using a range of strategies, including giving choices, accepting and understanding their challenges and using humour [16]. While this study provides insight into some barriers and facilitators to conversation for PwP, only PwP were interviewed. As the success of a conversation involves both the CP and PwP, there is a need to explore both the views of PwP and the CPs on what facilitators are. Furthermore, while some facilitators were described, the focus of this paper was not on facilitators to conversational participation. For example, PwP spoke about how to overcome the challenges of conversations generally; specific examples were not given.

Only one previously published study specifically investigated the facilitators and barriers to conversational participation [13]. In this study, six PwP and their CCPs were interviewed both jointly and separately about the barriers and facilitators to conversation participation. Barriers to the PwP participating in conversations included fatigue, communication impairment, cognitive impairment, background noise, being in different rooms, impaired hearing, needing to focus on the mechanics of eating, being in a wheelchair, reduced motivation and reduced need to participate [13]. Facilitators supporting participation in conversations included the CP repeating content back to the PwP to check that they had understood what was being said—the CP repeating what the PwP had said so that others could hear, reducing background noise, facing each other, being close, PwP preparing what to say in advance, PwP writing down key words and the CP slowing down the conversation [13]. The study focussed predominantly on communicative changes experiences by PwP and the impacts of these on everyday conversations, rather than a detailed exploration of what may facilitate effective conversations.

Moving beyond documenting barriers to conversations, to consider what facilitates effective communication is an important direction for research. Understanding how other people overcome challenges is potentially valuable information that may be used to influence the behaviour of PwP and their supporters. There is growing awareness of the importance of supporting individuals to take an active role in addressing issues and solving problems experienced as part of a chronic health condition [21]. This requires individuals to believe that they have the capacity to exert some level of control over their own functioning, known as self-efficacy [22,23]. Bandura [22] identified four areas that support the development of self-efficacy belief: (a) performance accomplishments, (b) vicarious experiences, (c) verbal persuasion, and (d) emotional arousal. While (a) and (d) refer to direct individual experiences of the individual, (b) and (c) reflect the influence of the social environment in developing a sense of self-efficacy. Seeing how other people achieve success in a task is helpful in enabling people to believe that they have the ability to achieve success. Understanding more about the effective strategies used by others to support conversational participation is key to supporting self-efficacy [24].

There is, therefore, a need to understand the barriers and facilitators to participation in conversations, with a greater emphasis on facilitators that both PwPs and their CPs find useful in supporting conversational participation. This study aimed to extend the work of Johansson et al. [13] by exploring the perspectives of PwP and the communication partners on factors that influence the participation of the PwP in conversation. Specifically, this study aimed to understand more about what PwP and their CPs perceived both the barriers and the facilitators to conversational participation to be.

## 2. Materials and Methods

The current study used a qualitative, exploratory design in order to explore the lived experiences of PwP and their CCP [25]. The research explored participant perspectives of barriers to, and facilitators of, effective conversational participation for the PwP. Joint and individual semi-structured online interviews were conducted with PwP and their CCPs. This method was adapted from Johansson et al. [13]. Dyad interviews were conducted initially to capture the shared understanding from the dyad, enabling enable co-construction around the topic. Subsequent individual interviews enabled participants to speak with the researcher about topics they may not have wished to discuss with their CP present or where their opinion differed to their CP. Semi-structured interviews followed a topic guide (Appendix A). All interviews were conducted by author 2. Joint interviews took between 40–85 min, while individual interviews ranged from 20–60 min. Online interviews were used as the study took place during the COVID-19 pandemic.

### 2.1. Participants

Convenience sampling was used to recruit eight dyads from a Parkinson’s research centre participant database and support groups for PwP in an Australian city. Each dyad consisted of a PwP and a close communication partner. Interested participants were given an information sheet and provided written consent. Participants were required to have internet access to enable online interviewing. Participants with Parkinson’s were required to be able to speak fluent English and to have a close relationship with a CP willing to participate in the study, a self-reported diagnosis of Parkinson’s for a minimum of 1 year, and a score of over 24 on the Modified Telephone Interview for Cognitive Status (TICS-M) [26,27]. Close communication partners were nominated by the participant with Parkinson’s and were required to be able to speak fluent English, have a close relationship with a person with Parkinson’s who had agreed to participate in the study, and to have known the person with Parkinson’s for more than 6 months.

PwP participants consisted of seven males and one female with a mean age of 72 years. CCPs consisted of one male and seven females with a mean age of 69 years. Table 1 provides participant demographic information.

### 2.2. Procedure

Participants were interviewed online using Cisco Webex, a videoconferencing application. Participants were at home when interviewed, with no third parties present in the room. Interviews were audio and video recorded and transcribed verbatim. Participants with Parkinson’s were offered the opportunity to break up interviews over time to accommodate self-reported fatigue and off periods linked to medication use. This was not requested by any participant.

### 2.3. Trustworthiness and Rigour

The Consolidated Criteria for Reporting Qualitative Research (COREQ) checklist guided the design and conduct of the research to promote rigour and trustworthiness [28]. Processes to support reflexivity were used to clarify potential biases brought to the research process [29,30,31]. Prior to data collection, the researchers developed a reflexivity statement and used reflective journaling and field notes to make thought processes and assumptions explicit [30,32]. These processes assisted researchers in attempting to explicitly frame presumptions and judgements held by the researchers during the analysis phase [33]. Member checking was not used in this project, as analysis was an interpretive process requiring abstraction and thematic construction by the researcher. As significant time had passed between the time of the interview and the writing of results, it was acknowledged that participants may be at a different phase of their experience, and the cognitive and caring challenges of prolonged engagement may affect participants [34]. Rigorous verbal checking/confirmation was within interviews to clarify key ideas expressed. Future research may seek to explore this topic using inclusive research methodologies.

### 2.4. Data Analysis

NVIVO software (version 12.5.0.815) was used to code data. Data were analysed using thematic analysis, with theme development based upon the process described by Vaismoradi et al. [31]. These four stages, described below, were used to systematically organise and analyse coded data and to abstract key ideas embedded within the data into themes [31]. Interviewing and data analysis occurred concurrently to capture implicit observations from reflections and to shape questioning in future interviews.

#### 2.4.1. Initialisation

Preliminary notes were taken to begin to capture key and recurrent ideas in the data.

#### 2.4.2. Construction

Coding used an inductive approach, with a subset of data collaboratively coded by two team members to support consistency of coding. A codebook defined and provided examples of each code to support internal coding consistency [35]. The codebook was repeatedly refined as part of an iterative process. An audit trail tracked coding decisions, alterations and amendments made during each coding session to increase transparency and trustworthiness during the analysis [31,32,35].

#### 2.4.3. Verification

Codes were clustered according to the element they described. Preliminary themes were repeatedly reviewed, refined and merged collaboratively by two researchers (authors 1 and 2).

#### 2.4.4. Finalisation

Themes and relationships between themes were finalised collaboratively with the larger research team.

## 3. Results

Participants described factors influencing communication participation that encompassed five themes, including factors related to the person with Parkinson’s, factors related to the CP, the complex and dynamic nature of conversations, the communication environment and the impact of experience in shaping participation in conversation. These elements formed the basis for conceptualisation of the themes (Figure 1).

Participants discussed both facilitators and barriers to conversation within four of the themes (person with Parkinson’s, communication partner, dynamic nature of conversion and communication environment). This facilitator and barrier notion framed subthemes. One theme (impact of experience) included subthemes not specifically representing either barriers or facilitators but describing the influence of experience in shaping participation in conversation. Themes and subthemes are listed in Table 2 and described in turn, with illustrative quotes highlighting salient points.


*Theme 1. Conversations are impacted by the skills and abilities of the person with Parkinson’s*


Changes in the abilities, communication, cognitive and motor skills of the PwP were described as impacting conversational participation. There were four subthemes which included how these changes acted as barriers to conversation and two including ways participants tried to address these issues to support conversation.


*Barrier 1.1 Changes in speech and language skills*


PwP and their CPs described a range of changes in their speech and language skills associated with Parkinson’s as impacting their ability to engage within conversations. These included issues with *speech production*, such as articulating clearly, having a soft, monotone voice, getting tongue-tied, stuttering, slurring, mumbling and having difficulty initiating speech; issues with *rate of speech* such as a faster or slower speech rate, pausing and delays in responses were described as impacting conversations; and issues with *language* skills such as short responses, the use of non-specific language, word-finding difficulties and difficulties finishing sentences.


*Barrier 1.2 Changes in body language and facial expression*


CPs described the impact of the PwP having a masked face, limited facial expression and reduced vocal expression on conversational interaction.


*It’s the lack of emotional connection which happens because of the masked face and the monotone. He may be feeling excited, unhappy, anything underneath and I can’t tell what it is. So, I can’t connect with his emotions.*
(CP2)


*Barrier 1.3 Changes in cognitive abilities*


The PWP’s cognitive abilities were frequently described by both PwP and CPs as impacting on conversations. PwP spoke of losing their train of thought, struggling to multitask (converse while performing any task) and “zoning out” of conversations.


*I don’t get in quick enough to … the spaces of the conversation. … you think of something really good to say but then you, then you forget it again.*
(PwP1)


*Barrier 1.4 External influences on skills and abilities*


Participants with Parkinson’s spoke about having variable conversational success linked to the use of substances, such as Parkinson’s medications and alcohol intake. Drinking alcohol was described at decreasing PwP intelligibility, while medication cycles impacted speech production and fatigue levels.


*Facilitator 1.1 PwP attempt to change the ways they communicate to improve participation in the conversation*


Most participants with Parkinson’s described the ways they attempted to alter the way they produced speech to improve their conversational success. This included speaking slower or louder, repeating themselves, making a conscious effort to articulate better and using short responses. Participants who spoke about dysfluency spoke of taking a breath and starting again to improve their ability to participate in conversation.

PwPs also described using specific non-verbal strategies to support their success in conversation such as raising their hand to take a turn in conversation, maintaining eye contact and reading body language. Other communication strategies such as replaying the CP’s question in their mind and visualising the words they wanted to say helped PwP to remember their message. Written communication such as emails were discussed by many dyads as being a vital strategy, as this mode enabled the person with Parkinson’s time to think about what they wanted to say.


*That’s a tactic he uses as well because his verbal communication can be so “unreliable” for the lack of a better word. He will spend a long time writing that email, for example. It won’t look like that when you read it, but he must’ve spent ages doing it. But he prefers that because then there’s no rush, he can think about what he wants to say.*
(CP2)


*Facilitator 1.2 CP prompts PwP to do something differently to improve participation in the conversation*


CPs also appeared to take an active role in giving feedback on the PwP’s speech production. CPs described asking the person with Parkinson’s to speak loudly or slowly, or to repeat parts of the message that were not understood. CPs would be explicit in prompting their partner to change something about their communication. CPs spoke of providing their partner with verbal prompts, such as suggesting questions they could ask of others to increase conversation length, reminding their partner of what they were talking about when required and suggesting words when having word-finding difficulties.


*He was reading an article from a newspaper, and it was a very lengthy article … and suddenly he was getting soft, and that’s when I said “you really need to raise your voice, I can hardly hear you”. And so, he raised it and it was fine.*
(CP5)


*My daughter tries to, … encourage me to … have a longer conversation, and say to me ‘Hi [name], how are you doing’ so I say ‘I’m fine’ would be my response. But she will say ‘then ask her back, how was your day?’ you know, so things like that. That’s the other way of extending the conversation.*
(PwP8)


*Theme 2. Conversations are impacted by the knowledge and behaviours of the CP*


The knowledge and behaviours of the CPs were also described by participants as important in influencing conversational participation. These are described in four subthemes: two related to barriers and two to facilitation of conversation.


*Barrier 2.1 Communication characteristics and behaviour of CP*


PwP indicated that the communication characteristics and behaviours of their CP, such as their hearing abilities and speech rate, impacted their ability to participate in conversations. Other communicative behaviours limiting conversational participation included dominating-type behaviours such as taking over conversation, not allowing the person with Parkinson’s an opportunity to take a turn to speak and speaking for the person with Parkinson’s. PwP and CPs also described how rapid topic changes made it challenging to keep up with the conversation.


*I tend to talk about something and then I’ll swap lanes and just expect PwP to know I’m now talking about this. And he says to me, “so we’ve changed subjects have we?” I said “yeah.” That’s probably something I’ve probably always done, but it’s probably harder for PwP to change lanes as quick as me now.*
(CP6)


*Barrier 2.2 Understanding and expectations by CPs*


Limitations in knowledge of the CP about how Parkinson’s-associated symptoms may impact conversations was a common idea represented by participants. Most dyads also described differing expectations in how much conversation they desired or expected from each other, with conversational partners often appearing to expect more interaction.


*Interviewer: Is there anything else that you think makes (conversations)really hard?*

*PwP: … Sometimes I don’t have anything to say.*
(PwP1)


*He doesn’t need people. He doesn’t need me talking really.*
(CP1)


*Facilitator 2.1 CP changes own communication style to support conversation*


Separate to the strategies enacted by the person with Parkinson’s, CPs also described how they attempted to change their usual communication style to support conversation. Changes described by CPs included repeating themselves, raising their voice and using more closed questions. CPs reported asking their partner specific questions to check that they heard and comprehended the message. Explicit verbal prompts such as telling their partner that they changed topics were considered useful in keeping conversations going.


*Now I say “on another matter” or “changing subject” [when changing topics]*
(CP6)

CPs also spoke about the value of written communication in facilitating conversations with PwP. Using a shared, paper-based diary schedule and written information was described as helpful in supporting conversation topics and memory. This strategy was described as providing the person with Parkinson’s with visual support and minimising confusion.


*Yep, that’d be one of the best things we’ve done, really [sharing a diary]. You’ve just got to remember to take it with you when you go out, because it stops the confusion. It’s in black and white so he can see it.*
(CP1)

CPs would also consciously prime their partner for a two-way conversation by asking them a question about an upcoming topic.


*I can ask him a question, rather than tell him something, so asking questions gets him in the zone.*
(CP1)

Many participants also spoke about how the CP would verbally open the floor to invite their partner to join a group conversation with other people and how much this facilitated participation.


*[PwP’s son] was talking about computers, and I was like “your dad’s got an interesting story about the internet with moving” and then PwP would take over. [PwP’s son]… actually listened, so that was nice, but sometimes… I sort of just open up the pathway.*
(CP6)


*Facilitator 2.2 CP understands communication challenges of PwP*


The importance of developing an understanding of the person with Parkinson’s communication challenges was a common theme. A CP spoke about attending support groups and speaking to other people in similar situations to find out how they managed their communication with their partners to prepare for challenges they may face as the disease progresses.


*I think the other thing is also knowing–I mean, some things we can’t change, [their condition], but knowing that this is going to come… I’m trying to start early by looking at other people’s situation and then sort of take it early. So, this whole year we spent a lot of time… this is to come, let’s prepare.*
(CP8)


*Facilitator 2.3 PwP tells CP what they need and what they are experiencing*


One method reported to facilitate a successful conversation was the PwP, informing the person they were conversing with what they needed when experiencing a communication challenge. However, several PwP said that they considered this information private and would only feel comfortable sharing this type of information with people close to them. Asking a CP to repeat the specific part of the message that was not understood and asking a CP to speak louder or slower were described as helpful by PwP when experiencing a communication challenge.


*I usually tell the other person what I’m experiencing. I’ve got Parkinson’s Disease. I suffer from short term memory loss. Sometimes the word disappears and comes back after 5 min, 10 min, half an hour. It’s happening more often and for longer periods, gradually progressing*
(PwP7)


*Theme 3. Conversations are impacted by the inherent complex requirements of the conversational exchanges*


Participants described barriers to conversation that related directly to the complex task requirements of conversation (three subthemes). They also discussed a range of strategies to explicitly accommodate for some of these complexities (five subthemes).


*Barrier 3.1 Conversations take energy*


PwP spoke of the need for energy and attention to maintain conversation and the impact of their low energy levels. Communication partners also spoke of feeling drained or too busy to converse with their partner, particularly with the additional roles they undertook to support their partner. Long conversations were described as particularly challenging.


*That was a pretty thorough conversation. I felt, I felt really tired at the end… Like I’d been doing physical exercise.*
(PwP7)


*Barrier 3.2 Conversations should be (but are not) natural*


Participants described diminished spontaneity of conversations and an unnatural flow owing to additional pausing and increased processing time required.


*CP: It’s more so [PwP] doesn’t have to… have a long-winded answer, so he can just give me a ‘yes’ or ‘no’ and not have to think about it too much.*

*PwP: Yes.*

*CP: Yeah. Although, if we have too many communications like that, it feels very… business-like.*
(Joint interview, Dyad 2)


*Barrier 3.3 Conversations are fast moving*


Participants frequently described a lack of opportunities for the PwP to participate in conversations due to being spoken over or left out of conversations, as well as “not being able to get a word in” as the conversation moved too quickly.


*He won’t have said one word because nobody sort of says, “how are you [PwP]?” and [PwP] might say “I’m good thanks” if there’s no one speaking but invariably, someone jumps in. It just seems to always happen. And so [PwP] doesn’t even get a chance to say, “I’m okay” or “have you seen the football?” or something. … So, there’s not a lot you can do about it, you know because you don’t want to be rude and say … um, “let’s give [PwP] a bit of time,”*
(CP1)


*Facilitator 3.1 Using strategies to meet the energy (and attentional) demands of conversation*


Ways in which the PwP managed their energy required for conversation reportedly enhanced conversational ability to participate in conversations. Managing fatigue by sitting down and resting before or after social events was considered useful. Participants spoke of managing the timing of their medications, reducing alcohol intake and taking caffeine supplements to assist with meeting and managing the energy demands of conversations.


*Facilitator 3.2 CP modifies the conversation to support participation by PwP*


Participants spoke about modifying the conversation to address competing demands. CPs discussed keeping their language simple, discussing one topic at a time, allowing the person with Parkinson’s additional time to respond, and getting their attention before starting a conversation. PwP described asking their CP to wait until their current task was completed before participating in conversation.


*Telling him things, at an appropriate time and (in a) space that’s he’s comfortable in and recognising that he needs to have my full attention. Um, so yeah, he needs to be available to listen to me, too.*
(CP1)


*Facilitator 3.3 Dyad strategies about how to manage specific communication challenges inherent in conversations*


Dyads spoke of planning topics of upcoming conversation as being helpful, such as finding common ground that people might want to talk about. Many dyads developed explicit plans for how they would manage conversations at social events.


*We discussed before we went [to a party] … that I would seek refuge under [CP]’s wing…every now and then but I would go out and…try and battle on my own…. So, …we did that. And I got along pretty fine. I was speaking quite well. When I got there, after an hour or so, my speech improved. And I ended up having conversations with three or four different people. When I got tired, I went back to [CP].*
(PwP2)


*Facilitator 3.4 Dyad uses non-verbal strategies to support participation in conversation by PwP*


The collaborative planning of strategies to support the PwP’s ability to participate in conversation, both with the close CP and in other social conversations, was a clear theme within the data. Using non-verbal strategies, such as maintaining eye contact and reading facial expressions, was a common theme for facilitating conversation, as was recognizing the importance of non-verbal modes of communication such as email for the PwP.

*When I am in a round table (discussion), conversation or party or afternoon tea or something, I often get talked over or… people don’t seem to even hear… with this whispery voice… I am trying a new technique of putting my hand up. See, when I, when I go to speak, up until now, I started speaking, somebody else started at the same second and I get passed over*.(PwP7)


*Facilitator 3.5 Planning the timing of conversations*


Planning the timing of conversations, such as not having conversations late at night and jointly deciding when to have particular conversations, was also described as useful.

*And I suppose late at night, tired, just leave it till the morning. Don’t talk about stressful stuff late at night, leave it for tomorrow (laughs)*.(CP6)

Participants also spoke about allowing time to revisit conversations to increase understanding. This included talking about their conversational challenges and strategies after being interviewed jointly in this research and revisiting conversations that occurred in large social groups once the dyad was home.


*Theme 4. Conversations are impacted by the contexts in which they occur*


The importance of the context in which conversations occurred, including the environment and the type of conversation, was discussed by participants. Strategies to address these challenges posed within the environment were described in four subthemes.


*Barrier 4.1 Environment*


The impact of the physical environment, particularly background noise, on the ability to comprehend, receive and produce a message were frequently raised as an important issue for all participants. Similarly, every dyad spoke of the difficulty they had when having conversations from different rooms and the importance of speaking face-to-face.


*It’s the conversations on the go, when you are in the next room, that don’t work*
(CP7)

Barriers were also reported in relation to the other elements of the conversational environment, such as the size of the group participating in the conversation. Participants found that conversations in larger groups presented issues such as being spoken over and being excluded from conversation. Smaller groups reported different challenges such as the person with Parkinson’s being required to respond more often, requiring more energy.


*Barrier 4.2 Activities*


Specific conversational activities such as phone calls and public speaking were considered difficult, as PwP had difficulty making themselves heard clearly and being confident in communicating. Other contexts requiring multi-tasking such as standing while speaking or holding a conversation while driving were also raised as challenging. Mealtimes were described as contexts where conversations were limited, as the participant’s attention and cognition was occupied by the mechanics of eating, drinking and swallowing.


*Facilitator 4.1 Planning or influencing size of the group*


Participants described managing the communication environment to facilitate conversations. The size of the group clearly influenced participation in conversation. Some participants spoke about enjoying taking the listener role when in larger groups. Many participants spoke about how much easier conversations with small groups than one-on-one conversations, as they were provided an opportunity to contribute without the pressure of being the only responder.

*For PwP, in the small groups where everyone else is talking, and you can just, just eat and listen…he finds that more enjoyable*.(CP2)


*Facilitator 4.2 Managing noise*


Participants discussed how managing noise also facilitated conversation. Dyads mentioned that they would often entertain at home, rather than in the community to control noise levels. Many participants spoke about choosing restaurants based on the environment and going out for earlier dinners to minimise background noise during conversation.


*In a restaurant, if it’s carpeted, there’s less background noise, it doesn’t reverberate. If there’s soft furnishings on the seat, rather than timber or metal, then the fabric absorbs the sound. If there’s drapes, it helps the sound.*
(CP5)


*Facilitator 4.3 Changing proximity*


The proximity between communication partners was reported to impact communication success. Participants spoke about asking their partner to come into the same room to speak face-to-face as being an important facilitator.


*Facilitator 4.4 Being comfortable with CP*


Participants discussed the importance of feeling comfortable with their CP to increase their willingness to participate in conversations. This included speaking to another person with Parkinson’s or being familiar with the person they are speaking with.


*Theme 5 Previous experience impacts conversation*


Participants discussed previous experiences that shaped the willingness and motivation to communicate. Three subthemes described how previous experiences impacted conversational participation.


*Subtheme 5.1. Previous success or failure in conversation shapes attempts to converse*


Previous success or failure in conversational exchanges reportedly shaped the PwP’s attempts to converse. Participants spoke about being aware that they did not ask their CP to repeat when they had not understood or heard. Previous failure in these attempts at conversational repair influenced the willingness of the PwP to continue to try to repair in these ways.


*After about … asking someone to repeat something maybe twice, you’ll get loathe to do it a third time, you nod your head and think let’s move on ‘cause, I’m never gonna get this one.*
(PwP5)


*Subtheme 5.2. Communication breakdown has emotional impact that impacts conversation*


The emotional impact of communication breakdown appeared to influence participation in subsequent conversations. Emotional reactions including anger, impatience and frustration were considered by both CPs and PwP as barriers to the desire to have or continue a conversation. Willingness to initiate conversations appeared to be influenced by loss of self-confidence from repeated conversation breakdowns.

Additionally, how previous conversational interactions were managed was raised by participants as impacting subsequent conversations. CPs in particular reported emotional reactions. Feelings of frustration, irritation and impatience were described as negatively impacting their willingness to continue a conversation with their partner.


*We will at times get a bit irritated with each other, me more with her. … just the mechanics of keeping the conversation going and doing those small things… I mean our communication has been probably the best thing of our relationship … it’s just harder now.*
(CP7)


*Subtheme 5.3 Insight and self-awareness*


Some participants indicated that PwP were not always aware of communication challenges in the moments they occur, such as issues with communication breakdown or volume. CPs and PwP talked about the importance of awareness of communication issues for recognising how to facilitate effective communication and interaction with others.


*CP: (PwP) might not always be aware that there’s a breakdown, so it feels like the onus (is) on me to recognise, right he’s having flat spot at the moment, or an off period… I’ll quite often I’ll have to say “Are you up to talking about…”*

*PwP: Yeah… that’s very helpful.*
(Joint interview, Dyad 2)

## 4. Discussion

In this study, PwP and their CP described the challenges they face in everyday conversation and the range of strategies they use to improve conversations. The strategies included those that were enacted in the moment and those that required advanced planning for future conversations. The findings indicate that conversational success is dependent on the skills and abilities of the PwP (Theme 1), the communication behaviours of the CP (Theme 2), the inherent complex requirements of the conversational exchanges (Theme 3), the communication environments/contexts in which the conversation is taking place (Theme 4), and previous experience with conversations (Theme 5). These themes are each discussed in relation to previous research on this topic.

Participants discussed how changes to PwP’s speech, language, non-verbal skills, and cognitive abilities due to Parkinson’s impacted their ability to participate in conversations (Theme 1). They also identified other factors that could exacerbate these difficulties, such as timing of medication and the use of alcohol. They reported that conversational success could be achieved by the PwP attempting to compensate for the changes in speech, language, body language, facial expressions, and cognitive abilities. Some of the strategies included using written communication (such as emails) to allow the PwP time to structure their message and the CP providing their partner with verbal prompts (such as suggesting questions they could ask) to increase conversation length. This adds to the growing body of literature which indicated that fatigue, speech, language, voice, and cognitive difficulties associated with Parkinson’s impact conversations [13,14,15,16] and that modifications made by PwP, for example writing down key words, may result in more conversational success [13]. These findings suggest that any intervention targeting the skills of PwP aiming to improve conversational success needs to go beyond focusing on improving speech and voice. Any effective intervention needs to address non-verbal skills, language, and cognition. Furthermore, intervention(s) should aim for PwP to use a range of strategies which may support conversations and increase their understanding of what other factors may exacerbate difficulties with conversations.

In addition to the challenges associated with changes in communication of the PwP, participants asserted that the knowledge and skills of the CP played a key role in the success (or otherwise) of conversations (Theme 2). The natural communication style of the CP appeared to influence conversational success, with some characteristics of the CP (e.g., speaking too fast, hearing loss) acting as barriers to successful conversation. In response to this, CPs described facilitating conversations by consciously adapting their natural communication style to support the communication needs of the PwP. This was achieved using a range of strategies, including the CP explicitly stating that they were changing the topic of conversation (e.g., “on another matter…”), the CP verbally opening the floor and inviting their partner to join a conversation (e.g., “what do you think?”), and using paper-based notes to provide visual support and reduce confusion during interactions. These findings build on previous research which found that CPs could provide support to PwP by giving choices [16], repeating content back to the PwP to check they had understood what was being said, repeating what the PwP had said so that others could hear, and slowing down the conversation [13]. As well as the impact of the communication skills of the CP, participants indicated that an understanding of the needs of the PwP by the CP reportedly influenced conversational success. When the CP understood the nature of those challenges, they were able to reflect upon how they could adapt their own communication to support the PwP. This leads us to suggest that any intervention to improve conversations for PwP needs to increase the CP’s knowledge of the PwP’s communication needs and should aim to increase the strategies used by the CP to promote conversational success. It is clear that both the PwP and the CP play an important role in the success (or otherwise) of conversations.

In addition to factors relating to the PwP and to the CP, participants in this study reported that the fundamental nature of conversation made it a challenging form of communication (Theme 3). Conversations typically progress quickly and change direction regularly, requiring focus and energy to keep up with the flow of ideas and the sudden attentional shifts. This, coupled with the range of cognitive symptoms associated with PD, made conversation difficult and confronting, and it stunted the flow of information. While the nature of conversations will always be complex, PwP and their CPs in this study described a range of strategies they used to manage those complexities. These conversational planning strategies supported the PwP to participate in conversations (e.g., non-verbal signals for needing a turn, planning timing of conversations). Similar strategies were also reported by Johansson et al. [13], including preparing what to say in advance and facing each other during conversations. Beyond the conversational analyses studies which largely report on communication strategies enacted in the moment [11], this finding shows that PwP and their CPs also explicitly strategize about how to engage in conversations in a form of metacognitive planning. Understanding the experiences of PwP and their CPs in supporting successful conversations, and knowledge of the strategies perceived of as useful, is an important direction for future enquiry. Participants in this study were clear that the nature of conversation itself makes communication inherently challenging, but they indicate that they attempted to manage those challenges.

As well as the inherent nature of conversation presenting challenges, participants also described environments and activities that increased difficulties in conversations (Theme 4). These included features of the physical environment in which conversations took place (e.g., noise, number of people) and activities that necessitated conversation (e.g., phone calls). The challenges relating to noisy conversation environments were also described in Johansson et al. [13]. In this study, PwP and their CPs often pre-planned strategies that they would use to support conversational success and negotiate problematic conversational settings, such as a noisy restaurant. This included practical strategies such as determining the size of the group (limiting to smaller numbers), managing noise by sitting away from speakers in restaurants or pubs, reducing physical proximity between members of the group, seating group members so that they were facing one another, and ensuring that the PwP did not have to multitask (e.g., standing/talking vs. sitting/talking). This is in line with the findings of Johansson et al. [13], who reported that people would attempt to reduce background noise and communicate in closer proximity. It is clear that any intervention approach for PwP that targets conversations needs to explicitly promote the role of environment and context and must explore how practical strategies might help to address the barriers associated with the environment and context.

Much of the research on interventions that target communication difficulties experienced by PwP and their CPs has focused on speech intelligibility and the volume of the PwP’s voice (e.g., Lee Silverman Voice Treatment) [8]. The findings of the current study suggest that any successful intervention must extend beyond those issues relating to the speech and voice of the PwP, to consider contextual influences on communication [6,36]. Interventions need to increase knowledge and awareness of what factors determine conversational success and what practical strategies could be implemented by both the CP and the PwP prior to, and during, conversations. The role of the CP in understanding what strategies help conversation is of particular importance if the PwP has any cognitive issues, such as attention deficits or slowed speed of processing, which may impact their ability to recall and implement practical strategies. In instances where the PwP has such cognitive deficits, the CP will not only be required to implement their own strategies to support conversation but will also need to remind and prompt the PwP to apply their strategies to the conversation. This scenario is particularly challenging and places much greater onus on the CP to facilitate and support conversation.

The current study highlights the need to involve both the PwP and the CP in any conversation therapy. There is only one published intervention approach that targets conversations and involves PwP and their CPs, called CPT; however, there has been limited research exploring the use and efficacy of CP-training based interventions for PwP [3]. Another approach is to share strategies that PwP and CPs perceive to have led to successful conversations with other PwP and CPs [37]. This may be particularly important for those in the early stages of Parkinson’s before the frequency of conversation difficulties increases. By giving people ‘vicarious experiences’ in the early stages of Parkinson’s, this may lead to them being more likely to implement strategies when needed due to increased self-efficacy [21,22,23]. This becomes even more important when we consider that in our research, PwP and their CPs discussed the influence of previous experience in shaping how they attempt to have conversations, with communication breakdowns having ongoing emotional impact that appear to influence future attempts to converse. However, in this approach, it is the perceived experience of success rather than an objective measure of conversational success. It also assumes that all PwP and their CPs will benefit from the same strategies, which has not been systematically explored in the current body of research.

This study, along with those by Johansson et al. [13] and Miller et al. [16], provides people with useful information about the experiences of those with PD and their CPs in having conversations. However, the sample size was small, and the participants came from a specific geographical location and language group. The findings may not be applicable to all other people with PD and their CPs. The findings regarding barriers and facilitators were the views of those involved in the study, and specific measurements of whether these factors impacted conversational success were not undertaken. Further research is needed to measure the impact of specific strategies on conversational success in various scenarios and the efficacy of conversation interventions for PwP and their CPs.

## 5. Conclusions

This study reports on the experiences of PwP and their CPs having conversations. It provides us with important information about what PwP and their CP perceive as the facilitators and barriers while having conversations, which can be shared with others who have PD and their CPs. This information should be considered in the design of future interventions for communication difficulties. It highlights the importance of early intervention for conversations and the need for more research into what aids conversations between PwP and their CPs in order to improve their quality of life.

## Figures and Tables

**Figure 1 brainsci-12-00944-f001:**
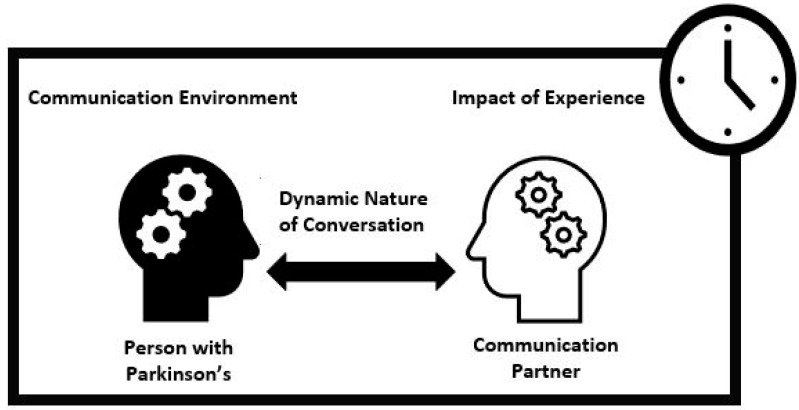
Conceptual diagram. Influences on conversation.

**Table 1 brainsci-12-00944-t001:** Summary of participant demographic information.

Dyad Number	Duration of Parkinson’s Diagnosis (Years)	Duration of Relationship (Years)	Age (Years)	
			Person with Parkinson’s	Communication Partner
1	15	49	69	66
2	4	42	61	60
3	4	53	74	74
4	5	58	84	82
5	5	55	77	77
6	2	8	70	62
7	21	34	77	65
8	1	40	66	65

**Table 2 brainsci-12-00944-t002:** Themes and subthemes.

Themes	Barrier Subthemes	Facilitator Subthemes
Conversations are impacted by the skills and abilities of the person with Parkinson’s.	1.1 Changes in speech and language skills 1.2 Changes in body language and facial expression 1.3 Changes in cognitive abilities 1.4 External influences on skills and abilities	1.1 PwP attempt to change the ways they communicate to improve participation in the conversation. 1.2 CP prompts PwP to do something differently to improve participation in the conversation.
2. Conversations are impacted by the knowledge and behaviours of the CP	2.1 Communication characteristics and behaviour of CP 2.2 Understanding and expectations by CPs	2.1 CP changes own communication style to support conversation 2.2 CP understands communication challenges of PwP 2.3 PwP tells CP what they need and what they are experiencing
3. Conversation is impacted by the inherent, complex requirements of conversational exchanges	3.1 Conversations take energy 3.2 Conversations should be (but are not) natural 3.3 Conversation are fast moving	3.1 Using strategies to meet the energy (& attentional) demands of conversation 3.2 CP modifies the conversation to support participation by PwP 3.3 Dyad strategies about how to manage specific communication challenges inherent in conversations 3.4 Dyad use in non-verbal strategies to support participation in conversation by PwP 3.5 Planning the timing of conversations
4. Conversations are impacted by the contexts in which they occur	4.1 Environment 4.2 Activities	4.1 Planning or influencing size of the group 4.2 Managing noise 4.3 Changing proximity 4.4 Being comfortable with CP
5. Previous experience impacts conversation	Subthemes	
	5.1 Previous success or failure in conversation shapes attempts to converse. 5.2 Communication breakdown has emotional impact that impacts conversation. 5.3 Insight and self-awareness	

Note. PwP, people with Parkinson’s; CP, communication partner.

## Data Availability

The data presented in this study are available on request from the corresponding author. The data are not publicly available due to ethical and privacy concerns.

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
