# Peer review of "Barriers and Facilitators to Conversation: A Qualitative Exploration of the Experiences of People with Parkinson’s and Their Close Communication Partners"

_brainsci, 2022, doi:10.3390/brainsci12070944_

Round 1
Reviewer 1 Report
In the present study, the Authors propose a novel approach to overcome the communication barriers that often PwP have to face. The outcomes of this study lay the groundwork for more effective approaches to improve the communication skills in communities in which PwP are present. The manuscript describes appropriately all the sections with scientific rigor.
Some minor text corrections are suggested.
Author Response
Thank yo for your review of this manuscript.
Response To Reviewers
Manuscript name: Barriers and Facilitators to Conversation: The Experiences of People with Parkinson’s and their Close Communication Partners
Date: 2 Jul 2022
Journal: Brain Sciences
Reviewer 1
Point 1:
In the present study, the Authors propose a novel approach to overcome the communication barriers that often PwP have to face. The outcomes of this study lay the groundwork for more effective approaches to improve the communication skills in communities in which PwP are present. The manuscript describes appropriately all the sections with scientific rigor.
Response 1:
Thank you for the positive comments about the overall direction of the research and the reporting of rigorous within the paper. Our team are strongly committed to building understanding of specific strategies for both people with Parkinson’s and their communication partners, incorporating the lived experience of people with Parkinson’s on their perspectives of what works in practice.
Point 2:
Some minor text corrections are suggested.
Response 2:
The reviewer suggests that there are minor text changes suggested in the review, however, no attachment was provided by the journal showing suggested text changes. It would be appreciated if the journal could provide this, as we are happy to review and amend it. We have emailed the editor about this.
Reviewer 2 Report
1. In the majority of the guidelines is recommended that the title provides the location where the study was done and the type of the study.
2. Check:
‘‘Communication Partner Training (CPT) '[see 17 for review]'.’’ – this could be confusing for the reviewer
3. In the reviewer's opinion, the investigation of Parkinson's disease severity as well as a depression scale would be mandatory to avoid possible confounding factors.
4. Did the authors request permission to perform ‘‘Modified Telephone Interview for Cognitive Status’’?
5. How do the authors analyze the low number of participants in this study?
6. IRB number should be provided.
7. The study provides an analysis between PwP and their CPs barriers and the facilitators to conversational participation. The majority of the information described is common knowledge in medical practice. Also, there are some studies specifically reviewing the barriers. What do the authors think that this study brings new to the literature?
Sakar BE, Isenkul ME, Sakar CO, Sertbas A, Gurgen F, Delil S, Apaydin H, Kursun O. Collection and analysis of a Parkinson speech dataset with multiple types of sound recordings. IEEE J Biomed Health Inform. 2013 Jul;17(4):828-34. DOI: 10.1109/JBHI.2013.2245674. PMID: 25055311.
Whitworth A, Lesser R, McKeith IA. Profiling conversation in Parkinson's disease with cognitive impairment. Aphasiology. 1999 Apr 1;13(4-5):407-25.
Author Response
Response To Reviewers
Manuscript name: Barriers and Facilitators to Conversation: The Experiences of People with Parkinson’s and their Close Communication Partners
Date: 2 Jul 2022
Journal: Brain Sciences
_________________________________
Thank you for your review of this manuscript which has assisted in refining the title, and explanations of the approach to research about conversations. Our specific comments are as follows:
Point 1. In the majority of the guidelines is recommended that the title provides the location where the study was done and the type of the study.
Response 1
Thank you for this feedback. We are not aware of any guidelines which stipulate this, however following on from this suggestion, we have modified the title to include the type of study. It is unclear why the location of the study would be needed in the title as it does not impact on interpretation and the findings of the study are unlikely to be influenced by the location of the study.
Suggested title: Barriers and Facilitators to Conversation: A Qualitative Exploration of the Experiences of People with Parkinson’s and their Close Communication Partners. This has been amended in the revised manuscript.
Point 2: ‘‘Communication Partner Training (CPT) '[see 17 for review]'.’’ – this could be confusing for the reviewer
Response 2
Communication Partner Training is a well-recognised approach to intervention in some areas of communication science. Ref 17 provides a review of partner training as an intervention approach for people with Parkinson’s. To simplify this for the reader, we now refer the this only as (17).
Point 3:
In the reviewer's opinion, the investigation of Parkinson's disease severity as well as a depression scale would be mandatory to avoid possible confounding factors.
Response 3
We have described this study as an early exploratory qualitative study, that aimed to explore the lived experiences of a range of people with Parkinson’s with self-identified communication difficulties. Narrowing the sample to a more homogenous group, by narrowing the severity range, or limiting to participants without depression, for example, may have limited the diversity we sought to explore within the study. We have described the duration of PD and other demographics of participants, which is an important strategy in showing rigour in qualitative research (Tong et al., 2007).
A framework of exploratory, descriptive, and comparative may be used to consider approaches to qualitative research (Rundle et al., 2019). This research is exploratory research which aims to capture a breadth of experiences. Capturing a diverse perspective is a commonly used approach when beginning to explore an under-explored phenomenon. The exploratory nature of this research project is because, to date, limited research has described how PwP and their communication partners attempt to respond to the barriers to conversation or draw on perspectives of the participants (lived experience) rather than observational, time-limited studies (conversational analysis). As the body of knowledge in this space grows, and we move towards more descriptive and comparative approaches, participant groups may become more homogenous, to deepen understanding of key concepts (identified in the exploratory phase) or to enable comparison.
This will be important to consider in later phases of research, as this study is an early component of a planned programme of research.
- Did the authors request permission to perform ‘‘Modified Telephone Interview for Cognitive Status’’?
Response 4.
The need to undertake a TICS interview was stipulated in the participant information sheet that participants were provided with before consenting. This process was clearly delineated and approved during ethical review and clearance.
- How do the authors analyze the low number of participants in this study?
Response 5.
Participant numbers are not analysed in qualitative studies. We sought to recruit 8 dryads and did so. This study involved 16 participants (8 dyads) and 24 interviews. This study used a small sample size which is typical of a qualitative study of this kind. This allows for a deep and detailed analysis that is central to the practice of qualitative enquiry. Unlike quantitive approaches, the sample size in qualitative research is often small, with a focus on meaning and its ultimate use (Sandelowski, 1995). Small sample sizes in interview-based research, reflective of numbers in this study have been shown to produce saturation (Hennink, Kaiser, & Marconi, 2017).
- IRB number should be provided.
Response 6.
This was removed for peer review. An identifiable copy of the manuscript is available that includes IRB details. In the non-blinded manuscript, this reads…Ethics approval for this study was granted by [removed here for blinding], ref. HRE2020-0256.
- The study provides an analysis between PwP and their CPs barriers and the facilitators to conversational participation. The majority of the information described is common knowledge in medical practice. Also, there are some studies specifically reviewing the barriers. What do the authors think that this study brings new to the literature?
Response 7.
Thank you for the opportunity to comment on this point, which we address in two parts.
A. Speech difficulties VS conversational participation: While the reviewer may see that this knowledge is “common in everyday medical practice”, this type of knowledge and detail about people’s experience and perspective in conversation is not well represented in the literature about PD. It maybe that the speech difficulties are common knowledge but this paper is about conversation difficulties which involve the interaction of two people and as our paper indicates are impacted by the PwP, their CP and the environment in which the conversation takes place.
The analytical detail presented in this paper is novel and is an important step in building a shared understanding, between clinicians, researchers and PwP, about the experiences of PwP and their CP. Anecdotally we are hearing from PwP that their communication partners are often not involved in communication therapy. Therapy seems to focus on making PwP louder and not address the dynamic interactions they have with their regular conversation partner. If the communication partner, the context and experience all play a role in the success or failure in conversation, then all of these factors should be addressed in communication therapy. It is important to represent the need to include communication partners in intervention and address contextual variables in the literature in creating debate about why interventions just focus on loudness/voice rather than all other factors.
B. Research approaches: Research on conversations by people with Parkinson’s and their partners can be grouped into two different approaches. The two approaches to research in the area were described in the manuscript, however, we see that we may not have been clear enough in contrasting these, so have amended the writing in section 3 to clarify this. [Lines 57-84].
Two approaches to understanding conversational challenges and facilities are used in the literature.
- Conversational analysis (observational). In this approach researchers collect recorded samples of conversation and analyse these for particular characteristics. This enables researchers to describe characteristics of discourse, and conversational breakdown and repair. It only captures a snapshot in time and does not consider the challenges of everyday scenarios and a range of contexts It does not harness the views of the speakers about their own challenges and successes in engaging in conversation. Additionally, participants are aware they are being recorded and this may mean their communication patterns may shift during data collection. There are more studies using this approach in PD research. The two papers offered by the reviewer are both observational studies, with the first focusing on describing speech impairments rather than the impact of these in conversation. They are therefore not relevant to the discussion of conversation difficulties and have not been added to our paper.
- Interview studies (lived experience). Interview studies do not aim to be an objective measure of communication behaviours and skills enacted in conversation. They aim to harness the views of the people who are having conversations about the unique challenges and successes they encounter in everyday conversations. Understanding the everyday experiences of participants is important in designing intervention approaches.
As the body of research in this space grows, ideally findings from both observational studies and lived experience studies will be synthesised to compare and explain findings, however, given there is little research specifically on the experiences of people with PD and their communication partners specifically in the field of conversation, the mandate for this type of research in building the evidence base is clear.
References
Hennink, M. M., Kaiser, B. N., & Marconi, V. C. (2017). Code saturation versus meaning saturation: how many interviews are enough?. Qualitative health research, 27(4), 591-608.
Rendle, K. A., Abramson, C. M., Garrett, S. B., Halley, M. C., & Dohan, D. (2019). Beyond exploratory: a tailored framework for designing and assessing qualitative health research. BMJ, 9(8), e030123.
Sandelowski, M. (1995). Sample size in qualitative research. Research in nursing & health, 18(2), 179-183.